# Molecular Characterization of *Staphylococcus aureus* Strains Isolated from Mobile Phones

**DOI:** 10.3390/microorganisms10030669

**Published:** 2022-03-21

**Authors:** Aída Hamdan-Partida, Samuel González-García, Francisco Javier Martínez-Ruíz, Miguel Ángel Zavala-Sánchez, Anaíd Bustos-Hamdan, Jaime Bustos-Martínez

**Affiliations:** 1Departamento de Atención a la Salud, Universidad Autónoma Metropolitana-Xochimilco, Mexico City 04960, Mexico; ahamp@correo.xoc.uam.mx (A.H.-P.); abuzimbel@hotmail.com (F.J.M.-R.); 2Doctorado en Ciencias Biológicas y de la Salud, Universidad Autónoma Metropolitana, Mexico City 04960, Mexico; samuel2023@hotmail.com; 3Departamento de Sistemas Biológicos, Universidad Autónoma Metropolitana-Xochimilco, Mexico City 04960, Mexico; mzavala@correo.xoc.uam.mx; 4Hospital Infantil de México Federico Gómez, Mexico City 06720, Mexico; anaidbustos@gmail.com

**Keywords:** mobile phones, students, *Staphylococcus aureus*, pharynx, nose, genotype, biofilm formation, PFGE, CA-MRSA, HA-MRSA

## Abstract

The widespread use of mobile phones (MP) among healthcare personnel might be considered as an important source of contamination. One of the most pathogenic bacteria to humans is *Staphylococcus aureus*, which can be transmitted through the constant use of MP. Nevertheless, which specific type of strains are transmitted and which are their sources have not been sufficiently studied. The aim of this study is to determine the source of contamination of MP and characterize the corresponding genotypic and phenotypic properties of the strains found. Nose, pharynx, and MP samples were taken from a group of health science students. We were able to determinate the clonality of the isolated strains by pulsed-field gel electrophoresis (PFGE) and *spa* gene typing (*spa-*type). Adhesin and toxin genes were detected, and the capacity of biofilm formation was determined. Several of the MP exhibited strains of *S. aureus* present in the nose and/or pharynx of their owners. methicillin-susceptible *Staphylococcus aureus* (MSSA), hospital-acquired methicillin-resistant *S. aureus* (HA-MRSA), and community-acquired methicillin-resistant *S. aureus* (CA-MRSA) strains were found, which indicated a variety of genotypes. This study concludes that MP can be contaminated with the strains of *S. aureus* present in the nose and/or pharynx of the owners; these strains can be of different types and there is no dominant genotype.

## 1. Introduction

Mobile phones are almost omnipresent and are necessary devices for healthcare workers. However, they have also become a source of contamination of nosocomial agents such as *Staphylococcus aureus* [1,2,3,4]. *S. aureus* is an important pathogen that causes a wide range of infections, ranging from mild skin infections to death [5,6]. Thus, *S. aureus* represents a major public health problem, especially in the case of Methicillin-resistant *S. aureus* (MRSA) strains, which are more pathogenic. This is the case for both hospital-acquired (HA-MRSA) and community-acquired (CA-MRSA) strains [7]. Consequently, contaminated MP represent a potential public health risk since they can be reservoirs for this pathogenic microorganism and allow its easy transmission.

Several studies have corroborated MP contamination in health personnel both in the intensive care unit (ICU) and other hospital areas [3,8,9,10], as well as in the MP of health science students [11,12,13].

MP contaminated with *S. aureus* permit the mobility of strains from the hospital to the community and from the community to the hospital. Even though the mobility of the strains has been reported [7,14], it is important know how it occurs.

For a long time, the nose has been considered as the main ecological niche of *S. aureus* [15]. However, there is evidence that it can colonize the skin, axillae, groin, rectum, hands, and pharynx [16,17,18,19]. Furthermore, recent studies demonstrated that the colonization of the pharynx by *S. aureus* can be greater than in the nose, which puts into question that the latter is the main colonization niche [19,20,21]. Therefore, nose and pharynx can also be a source of *S. aureus* MP contamination.

During the colonization process by *S. aureus*, binding to the host cell surface in a reversible or irreversible manner is mediated by the so-called microbial surface component, recognizing adhesive matrix molecules (MSCRAMM) [22].

Within the MSCRAMM are adhesins such as the fibronectin binding proteins (FnBPA and FnBPB), the serine-aspartate repeat protein family (SdrC, SdrD and SdrE), clumping factors (ClfA and ClfB), the collagen-binding adhesin (Cna), and protein A (Spa), among others. These proteins are associated with the process of binding to the host matrix, which initiates cell adhesion and/or biofilm development. ClfB, FnBP, and SdrC facilitate biofilm accumulation by promoting intercellular attachment soon after initial attachment [23].

Another efficient mechanism used by *S. aureus* to colonize is the formation of biofilm [24,25,26]. An important component in *S. aureus*, biofilm is the polysaccharide of intercellular adhesion (PIA), which accounts for most of the biofilm-forming extracellular matrix of staphylococci [27]. PIA synthesis is mediated by the *icaADBC* locus and is part of the accessory genes and not the bacterial genome, indicating that it is not found in all *S. aureus* strains. Its presence is observed exclusively as part of a plasmid of staphylococcal strains that form biofilms [28].

In addition, *S. aureus* produces a large amount of toxins that account, in a large proportion, for the resulting infections and damage that this microorganism produces [29]. Among the toxins produced by *S. aureus* are staphylococcal enterotoxins (SE), toxic shock syndrome toxin 1 (TSST-1), exfoliative toxins (ET), and Panton-Valentine leukocidin (PVL); these toxins are very important since they are implicated in food poisoning, toxic shock syndrome, scalded skin syndrome, and other diseases [29,30]. 

Adhesion and biofilm formation processes may be factors for this bacterium to remain in the MP. Likewise, the type of toxins carried by the strains is an important factor that must be determined in the isolated strains.

The aim of this study is to identify the sources of MP contamination with *Staphylococcus aureus* and to characterize the genotypic and phenotypic properties of the strains isolated from the MP.

## 2. Materials and Methods

### 2.1. Sample and Isolated Strains

Paired nasal and throat swabs were collected from 200 university health science students in the first year of their degree who are not yet working in hospitals. Of these, 33.5% (67) were men and 66.5% (133) women, both with a mean age of 22.1 years. The students’ mobile phones were swabbed with sterile cotton and the same tests for microbiological identification were applied. All participants provided their informed consent to participate as volunteers. No incentives were offered. The project was approved by the Ethics Committee of the Biological Sciences and Health Division of the UAM-Xochimilco (Document: DCBS.CD.056.18). The swabs were placed in soy trypticase broth overnight at 37 °C. Subsequently, the samples were seeded on mannitol salt agar and left at 37 °C for 24 h.

### 2.2. Microbiological and Biochemical Identification

Mannitol fermentation-positive isolates were further analyzed to identify *S. aureus* strains. We performed Gram stain, catalase, and coagulase tests on pure colonies, and used the API Staph system (bioMérieux, Mexico City, Mexico) for bacterial identification.

### 2.3. Methicillin Susceptibility Testing

The presence of methicillin-sensitive *S. aureus* (MSSA) or methicillin-resistant *S. aureus* (MRSA) strains was determined by determining MIC to oxacillin, according to CLSI procedures [31]. Strains were identified as MRSA if the MIC was ≥4 mg/mL. The *S. aureus* strain used as negative control was ATCC2913, while ATCC43300 was used as positive control.

### 2.4. Detection of mecA Gene 

The Wizard genomic DNA purification kit (Promega, Madison, WI, USA) was employed for bacterial DNA extraction, following the manufacturer’s instructions.

PCR assays were performed for *mecA* gene, utilizing primers and conditions as previously reported [32], using a MyCycler Thermocycler (Bio-Rad, Hercules, CA, USA). Amplicons were analyzed on 1% agarose gels stained with ethidium bromide. *S. aureus* ATCC43300 was the positive control.

### 2.5. Detection of Hospital-Acquired Methicillin Resistant Staphylococcus aureus (HA-MRSA) or Community-Acquired Methicillin-Resistant Staphylococcus aureus (CA-MRSA) 

The strains that possessed the staphylococcal chromosome cassette *mec* (SCC*mec*) type IV or V and the Panton-Valentine leukocidin (PVL) genes were classified as CA-MRSA, while the HA-MRSA strains carry SCC*mec* types I, II, or III and rarely possess PVL [7,33]. Determination of SCC*mec* was carried out by employing two types of previously described multiplex PCR [32,34]. The following *S. aureus* strains from the ATCC (BAA strains) and the Network of Antimicrobial Resistance in *Staphylococcus aureus* (NRS strains) collections were used as positive controls: BAA44 for SCC*mec* type I; BAA41 for SCC*mec* type II; BAA39 for SCC*mec* type III; NRS643 for SCC*mec* type IV; and NRS745 for SCC*mec* type V. 

The presence of PVL was determined by amplification of the lukS-PV/lukF-PV genes using the PCR [35]. Strain NRS213 was used as a positive control.

### 2.6. Typing with the spa Gene (spa-Typing)

Typing of *S. aureus* strains using the protein A gene (*spa*-typing), was obtained by amplifying the *spa* gene through PCR and subsequently sequencing the amplicons [36]. Likewise, *spa*-types were assigned using the SPA Searcher (available at http://seqtools.com accessed on 10 November 2021) and Ridom GmbH (available at http://spaserver.ridom.de/ accessed on 10 November 2021) websites.

### 2.7. Pulsed-Field Gel Electrophoresis (PFGE) Typing

The determination of the clonality of the isolated *S. aureus* strains was carried out by means of PFGE, the extraction of bacterial DNA and its digestion with the enzyme *SmaI* was carried out following the methodology previously described [37]. Samples were run on a CHEF-DR II system (Bio-Rad, USA). Gels were photographed and digitized using a Bio-Rad Gel Doc (Bio-Rad, USA). The band patterns obtained by PFGE were analyzed with Gene Directory and Gene Tools software (Syngene, Cambridge, UK). We applied the unweighted pair group with mathematical average (UPMGA) based on Dice coefficients to obtain the percent similarities. A band position tolerance of 1.25% was established. For strain typing we used the criteria described by Tenover et al. [38].

### 2.8. Detection of Toxin and Adhesin Genes

The toxin genes *sea*, *seb*, *see*, *etb*, and *tst*, as well as the adhesin genes *fnbA*, *fnbB*, *cna*, *clfA*, *clfB*; *icaA*, *icaD*, and *sdrC*, were detected in the strains of *S. aureus* isolated by PCR, as described previously [39,40,41]. The *S. aureus* strains that served as positive control in the PCR were NRS111 for *tst*, *sea*, and *see*, NRS123 for *can*, NRS266 for *etb* and *seb*, BAA1556 for *clfA* and *clfB*, and ATCC2913 for *fnbA*, *fnbB*, *icaA*, *icaD*, and *sdrC*.

### 2.9. Biofilm Analysis

Biofilm formation was observed for the isolated *S. aureus*, as described previously [42].

### 2.10. Statistical Analysis

We performed the corresponding descriptive analysis of the measures of central tendency and dispersion; and the categorical variables were expressed as a percentage. To establish the relationship between groups of carriers, the Chi-square test, the Fischer exact test, and the *Z* test were applied. SPSS Statistics 25.0 (IBM, Armonk, NY, USA) software was used to carry out the analysis. A value of *p* < 0.05 was considered as statistically significant.

## 3. Results

### 3.1. Detection of Staphylococcus aureus Carriers

Of the 200 students analyzed in parallel, we found that 53% (106) were carriers of *S. aureus.* A higher percentage of exclusive pharynx carriers (25.5%) was observed, compared to exclusive carriers of the nose (15%) (*p* = 0.025). It should be noted that 12.5% were nose and pharynx carriers at the same time, Table 1.

When grouped by gender, analysis showed that 71.6% of men and 43.6% of women carried *S. aureus* at one or more of the sites analyzed. Therefore, the presence of *S. aureus* is greater in men than in women (*p* = 0.008), Table 1.

However, when the prevalence of *S. aureus* was analyzed comparing by gender and isolation sites, we found that for the pharynx, men presented a prevalence of 26.8% and women of 24.8%. In the case of the nose, the prevalence in men was 23.8% and in women 10.5%. Finally, the prevalence for men and women in both sites was 20.8% and 8.2%, respectively. No statistically significant differences were found between men and women in any of the cases (*p* > 0.05), Table 1.

Only 17% of the students presented contamination both in the nose and/or pharynx and their MP. 

Moreover, only 19 mobile phones (9.5%) of the total were contaminated with *S. aureus* (Table 1). The owners of 10 of them were also carriers of *S. aureus* on the nose and pharynx; four were exclusive carriers on the nose and the same number were exclusive pharynx carriers, while only one student presented MP contamination without being a carrier either on nose or pharynx.

### 3.2. Characterization of Isolated Staphylococcus aureus Strains

Overall, 144 strains of *S. aureus* were isolated: most of them were MSSA (76.4%) and only 23.6% were MRSA, so there is a greater colonization with MSSA strains (*p* < 0.001). Only the strains isolated from students with contaminated mobile phone were characterized further. Thus, a total of 47 strains were studied (14/nose, 14/pharynx, and 19/MP). Our study shows that 61.7% were MSSA strains, 19.1% HA-MRSA, and 17% CA-MRSA. Only one of these strains did not possess the PVL gene (despite having SCC*mec* IV); consequently, it could not be classified and it was only reported as MRSA (Figure 1).

Of the strains of *S. aureus* found in MP (19 isolates), by PFGE it was established that nine of them were identical to the strains carried by the owners of the MP, six students had it on the nose (students 17, 34, 160, 167, 191, and 192), and three were simultaneous nose and pharynx carriers (students 2, 85, and 88), as shown in Figure 1. In addition, eight of the strains found in the MP were strains clonally related with respect to strains found in other carriers and only two strains were different from the total of the strains analyzed (132M and 149M) (Figure 1). Likewise, we found strains of multiple *spa*-types; the most recurrent were *spa*-types *t-012* and *t-189*. Finally, 11 of the strains were identified as MSSA (57.9%), five as HA-MRSA (26.3%), and three as CA-MRSA (15.8%) (Figure 1).

### 3.3. Genotyping of the Strains Isolated in the MP

Table 2 shows the genes of toxins and adhesins detected in the *S. aureus* strains isolated in the carriers of contaminated MP. Multiple patterns were observed and strains belonging to the same PFGE group can differ in their toxin/adhesin content.

Among the adhesins found, the most abundant gene was *icaD* (100%), followed by *clfB* (82.9%), *cna* (70.2%), *sdrC* (80.8%), *fnbB* (46.8%), and *clfA* (42.6%). The least present genes were *icaA* (38.2%) and *fnbA* (27.7%) (Table 3).

The predominant toxin genes found were the *seb* gene (63.8%), with a greater presence in the strains isolated from the pharynx (*p* = 0.036) and the *tst* gene (57.8%), with a greater presence in nose and MP strains (*p* = 0.046), followed by the *sea* gene (29.8%) and the PVL gene (19.1%). It should be mentioned that the *see* and *etb* genes were not found (Table 3).

We identified various genogroups in the strains isolated in the MP (Figure 2); the most abundant genogroup contains five of the genes detected: *icaD*, *icaA*, *clfB*, *can*, and *seb*. Regarding toxins, the most abundant genogroup contained the *seb* and *tst* genes. The *sea* and PVL genes were included in the other genogroups. In the case of adhesins, the most abundant genogroup contains the *icaD* and *fnbB* genes, closely followed by the genogroup that contains the *icaD* and *sdrC* genes.

### 3.4. Biofilm Formation of S. aureus Strains Isolated from MP

In total, 46.8% of the strains isolated from contaminated MP carriers with *S. aureus* presented a weak biofilm formation, 29.8% presented strong biofilm formation, and 23.4% had moderate biofilm formation. The same pattern was observed in the strains isolated from MP: the majority presented a weak biofilm formation (63.2%), 26.3% showed strong biofilm formation, and only 10.5% presented a moderate biofilm formation (Table 2). No direct relationship was observed between the pattern of adhesins presented by the strains and the type of biofilm that was formed (Table 2).

## 4. Discussion

Contamination of MP with *S. aureus* poses a health risk, especially when the carriers of the devices are healthcare personnel. In this case, *S. aureus* could be spread to patients or to various healthcare center areas, causing nosocomial infections and other detrimental effects [2,3,4,43,44]. 

In this work, the contamination by *S. aureus* of health sciences students’ MP was studied. In a manner consistent with similar studies, we found MP contaminated by *S. aureus* [11,12,45,46]. 

The percentage of contamination with *S. aureus* of the MP of the group of 1st-year health science students whom we analyzed was lower than the one registered by another study (77.8%) [11]. In contrast, we only found a percentage of contamination of 9.5%: this finding is more consistent with other studies that registered lower contamination ranging from 3.4% to 16.2% [12,13,45]. These differences in the percentages of contamination might be due to various factors, especially environmental ones. Recent research demonstrates that the contamination percentage of MP is higher among healthcare workers, when compared to people who do not work in hospitals [47]. Furthermore, a greater incidence of contamination of MP has been found in studies carried out in developing countries, when compared to developed countries [48]. 

As reported by other research projects, our study shows that the main *S. aureus* colonization site was the pharynx rather than the nose [19,20,21]. Therefore, the pharynx is an important ecological niche to study for eventually controlling this bacterium.

The fact that we observed both MSSA and MRSA accounts for the diversity of circulating strains in this population. It was possible to verify by PFGE and *spa* typing that the *S. aureus* strains that contaminate MP were also colonizing the owner of the MP either in the nose, or pharynx, or both. Consequently, we could confirm that there is a transmission from the pharynx of the host to the MP and not only through the nose [10] or the hands [10,12,18,49], as mentioned by other studies. The reason why not all the strains found in the MP were also found in the owners could possibly be the presence of bacteria on the hands, or that the contamination was from another source. However, almost all *S. aureus* strains are related within the analyzed population, except for two strains that differed from the majority (Figure 1). This could be explained by the fact that it is an open population and that the exchange of bacteria occurs on a large scale, which is confirmed by observing that there are many *sap*-types.

We were able to identify MSSA, HA-MRSA, and CA-MRSA strains, both in students and in their MP. Brady et al. suggest that there is a low possibility of MP contamination with MRSA strains in non-hospital environments [50]. Our results show that it is possible that MP are contaminated with MRSA strains (Figure 1), even though students are not in hospital settings or environments, as is the case of the students analyzed in this work.

Hopefully, the percentage of MRSA is low, as found in other studies [2,49,51]. Among the examined literature, only one study documented the presence of HA-MRSA and CA-MRSA strains in MP [52]. This implies that CA-MRSA strains can also spread through MP, just like other types of strains; as previously mentioned, cross-contamination between hospitals and the community and vice versa may occur [53,54,55].

The predominant adhesin genes identified in the studied strains of the students and their MP were *icaD*, *clfB*, *sdrC*, and *cna* (Table 2 and Table 3, Figure 2). Something similar was documented by Noumi et al., since they also found a high percentage of MP strains that presented the *icaD*, *can*, and *fnbA* genes [56]. Nevertheless, we found that the *fnbA* gene is present in a higher percentage in pharynx strains than in those isolated from MP (Table 3). 

All strains of *S. aureus* isolated from MP showed biofilm formation ranging from weak to strong, with weak-forming strains predominating (Table 2). However, we found no relationship between the adhesin genes present in the strains and weak, moderate, or strong biofilm formation (Table 2).

We could advance the hypothesis that both adhesins and the biofilm formation are important for the contamination of MP by *S. aureus*, as occurs in the colonization of humans.

Regarding our finding of toxin genes in the studied *S. aureus* strains, the *seb* and *tst* genes were the most abundant; the *seb* gene, which codes for the SEB enterotoxin, was found in a greater proportion in strains isolated from the pharynx, followed by the strains isolated from MP. In turn, the *tst* gene that codes for TSST-1 was found in a higher percentage in strains isolated from the nose and from MP (Table 3). Additionally, we identified MP strains carrying the *sea* and *lukS-PV/lukF-PV* genes. The presence of toxin-carrying *S. aureus* strains in the MP is important since these toxins could be spreading through these devices.

Finally, the high genetic variability prevailing in the *S. aureus* strains isolated from students and their MP (Table 2, Figure 2) is consistent with the evidence proposed by other studies; this variability is also observed among humans and MP [20,56]. This implies a high number of different *S. aureus* strains circulating in the environment, so epidemiological surveillance studies of *S. aureus* should continue to be carried out.

## 5. Conclusions

Our research found that health science students’ MP were contaminated with *Staphylococcus aureus*. Moreover, our study shows that MP contamination can be from *S. aureus* that colonize the pharynx or nose of MP owners. MP can be reservoirs for MSSA, HA-MRSA, and CA-MRSA strains. The strains of *S. aureus* isolated from the MP possess differentiated genes for adhesins and toxins. Thus, they present a high genotype diversity, with the ability to form biofilms. Consequently, it is necessary to implement hygiene measures to prevent MP from becoming a source of *S. aureus* contamination.

## Figures and Tables

**Figure 1 microorganisms-10-00669-f001:**
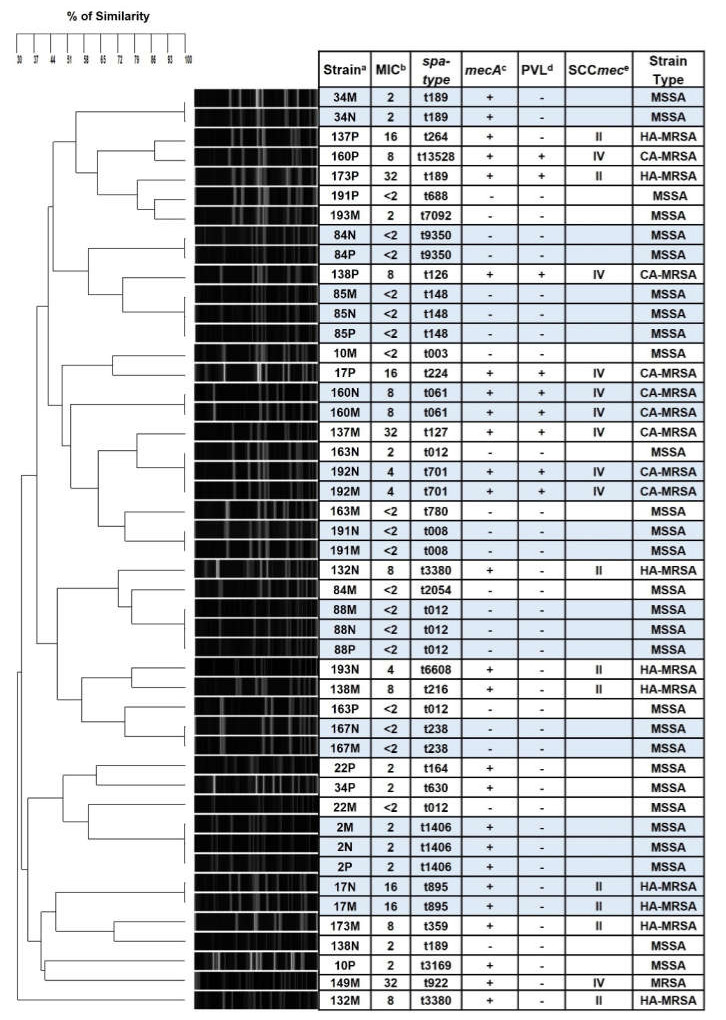
*SmaI*-PFGE dendrogram comparing *Staphylococcus aureus* strains isolated from carriers and their contaminated mobile phones. Carrier number and site, oxacillin MIC, *spa*-type, *mecA* and PVL genes detection, and SCC*mec* for each strain are shown. ^a^ Carrier number/site: M = mobile phone; P = pharynx; N = nose. ^b^ Oxacillin MIC in μg/mL. ^c^ – not detected; + detected. ^d^
*lukS-PV/lukF-PV* genes which encode PVL was detected, -not detected; + detected. ^e^ SCC*mec* type was tested only when the strain had MIC ≥ 4 μg/mL. Blue cells: strains from the same carrier that are identical by PFGE and *spa*-type.

**Figure 2 microorganisms-10-00669-f002:**
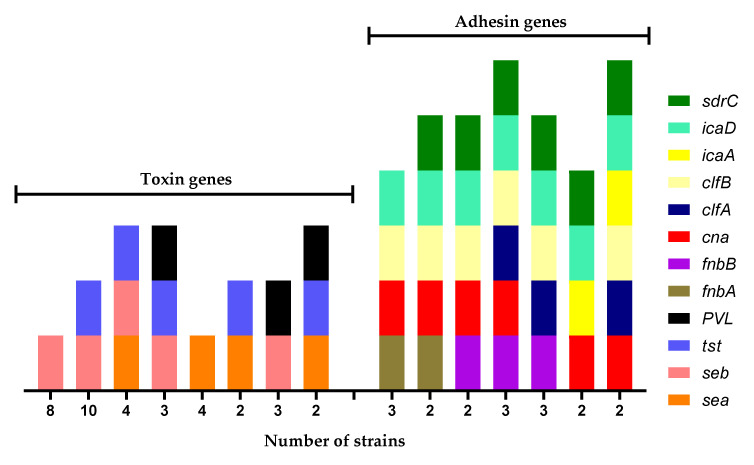
Genogroups of *Staphylococcus aureus* strains isolated from mobile phones.

**Table 1 microorganisms-10-00669-t001:** Distribution of *Staphylococcus aureus* strains in the different niches by gender.

	*n*	Pharynx or Nose	Pharynx	Nose	Pharynx and Nose	Mobile Phone
Men	67 (33.5%)	48 (71.6%) *	18 (26.8%)	16 (23.8%)	14 (20.8%)	7 (10.4%)
Women	133 (66.5%)	58 (43.6%)	33 (24.8%)	14 (10.5%)	11 (8.2%)	12 (9.0%)
Total	200 (100%)	106 (53.0%)	51 (25.5%)	30 (15.0%)	25 (12.5%)	19 (9.5%)

Chi-square test, Fischer’s exact test, and Z test were performed. * *p* = 0.008.

**Table 2 microorganisms-10-00669-t002:** Toxin genes, adhesin genes, and biofilm formation detected in *Staphylococcus aureus* strains isolated from students and their mobile phones.

	Toxin Genes ^b^				Adhesin Genes ^b^								Biofilm Foration ^c^
Strain ^a^	*sea*	*seb*	*tst*	*PVL*	*fnbA*	*fnbB*	*cna*	*clfA*	*clfB*	*icaA*	*icaD*	*sdrC*	
2N	−	+	−	−	+	−	+	−	+	−	+	−	+
2P	−	−	−	−	+	−	+	−	+	−	+	−	+
2M	+	−	−	−	+	−	+	−	+	−	+	−	+
10P	−	+	−	−	+	+	+	−	+	−	+	−	+
10M	−	+	−	−	+	−	+	+	+	−	+	−	++
17N	−	+	−	−	+	−	+	−	+	−	+	+	+
17P	−	+	−	+	+	−	+	−	+	−	+	+	+
17M	−	+	+	−	−	+	−	−	+	−	+	+	+
22P	−	−	−	−	+	+	−	+	+	−	+	+	+++
22M	+	−	−	−	+	+	+	+	+	−	+	+	+
34N	−	−	+	−	+	+	+	−	+	−	+	+	++
34P	−	+	+	−	+	−	+	−	−	−	+	−	+++
34M	−	−	−	−	−	+	+	−	+	−	+	+	+++
84N	−	−	−	−	−	+	+	+	+	−	+	+	+++
84P	−	+	+	−	−	+	+	+	+	−	+	+	+++
84M	−	+	+	−	−	+	+	+	+	−	+	+	++
85N	−	+	−	−	−	+	−	+	+	−	+	+	+++
85P	−	+	−	−	−	+	−	+	+	−	+	+	+++
85M	−	+	−	−	−	+	−	−	+	−	+	+	+++
88N	−	+	+	−	−	+	+	−	+	−	+	+	++
88P	−	+	+	−	−	−	+	−	+	−	+	+	+++
88M	−	+	+	−	−	−	+	−	+	−	+	+	++
132N	+	−	−	−	−	−	−	−	+	+	+	+	++
132M	+	−	−	−	−	−	−	−	+	−	+	+	+
137P	+	+	−	−	−	+	−	−	+	+	+	+	+
137M	+	−	+	+	−	−	−	+	+	−	+	+	+
138N	+	−	+	−	−	+	−	−	+	−	+	+	+++
138P	−	+	−	+	−	+	−	+	+	−	+	+	++
138M	−	+	+	−	−	−	−	+	+	−	+	+	+++
149M	−	+	−	−	−	−	+	+	+	−	+	+	+
160N	+	−	+	+	−	−	+	−	+	+	+	+	+
160P	−	+	−	+	−	+	+	−	−	+	+	+	+
160M	−	−	+	+	−	−	+	−	−	+	+	+	+
163N	−	+	+	−	−	−	+	−	−	+	+	+	+
163P	+	−	+	−	−	−	+	+	+	+	+	+	++
163M	−	−	+	−	−	−	+	+	−	+	+	+	+
167N	+	+	+	−	−	−	+	+	−	+	+	−	+
167M	+	+	+	−	−	−	+	−	+	+	+	−	+
173P	−	+	+	+	−	+	+	+	−	+	+	−	++
173M	+	+	+	−	−	−	+	+	+	+	+	+	+
191N	+	−	+	−	−	+	−	+	−	+	+	+	+
191P	−	+	−	−	+	−	+	+	+	+	+	+	++
191M	−	−	+	−	−	+	−	−	+	+	+	+	+
192N	−	+	+	+	−	+	+	−	+	+	+	+	++
192M	−	+	+	+	+	+	+	+	+	+	+	+	+++
193N	−	+	+	−	−	−	+	−	+	−	+	+	+++
193M	+	+	+	−	−	−	+	−	+	+	+	+	+++

^a^ M = mobile phone; N = nose; P = pharynx; ^b^ + = detected, − = not detected. Blue cells: identical strains by PFGE and *spa*-type. ^c^ + weakly biofilm formation; ++ moderately biofilm formation; +++ strongly biofilm formation. Cells with the same color group strains with the same pattern of adhesin or toxin genes.

**Table 3 microorganisms-10-00669-t003:** Toxin and adhesin genes in *Staphylococcus aureus* strains isolated from students and their mobile phones.

	Nose *n* = 14	Pharynx *n* = 14	Mobile Phone *n* = 19	Total *n* = 47
Toxins				
*sea*	5 (35.7%)	2 (14.2%)	7 (36.8%)	14 (29.8%)
*seb*	8 (57.0%)	11 (78.5%) *	11 (57.8%)	30 (63.8%)
*see*	0	0	0	0
*etb*	0	0	0	0
*tst*	9 (64.2%) *	5 (35.7%)	11 (57.8%) *	25 (53.1%)
*pvl*	2 (14.2%)	3 (21.4%)	4 (21.0%)	9 (19.1%)
Adhesins				
*fnbA*	3 (21.4%)	6 (42.8%)	4 (21.0%)	13 (27.7%)
*fnbB*	7 (50.0%)	8 (57.1%)	7 (36.8%)	22 (46.8%)
*cna*	10 (71.4%)	10 (71.4%)	13 (68.4%)	33 (70.2%)
*clfA*	4 (28.5%)	7 (50.0%) *	9 (47.3%) *	20 (42.6%)
*clfB*	11 (78.5%)	11 (78.5%)	17 (89.4%)	39 (82.9%)
*icaA*	6 (42.8%)	5 (35.7%)	7 (36.8%)	18 (38.2%)
*icaD*	14 (100%)	14 (100%)	19 (100%)	47 (100%)
*sdrC*	12 (85.7%)	10 (71.4%)	16 (84.2%)	38 (80.8%)

Chi-square test, Fischer’s exact test and Z test: * *p* < 0.05 were performed.

## Data Availability

Not applicable.

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
