# Peer review of "Molecular Characterization of Staphylococcus aureus Strains Isolated from Mobile Phones"

_microorganisms, 2022, doi:10.3390/microorganisms10030669_

Round 1

Reviewer 1 Report

The manuscript deals with an interesting topic in the field of epidemiology, concerning a transmission of S. aureus strains, including MRSA strains, through mobile phones. The authors have provided detailed phenotypic and genotypic characteristics of 47 S. aureus strains isolated from mobile phones  and their owners.

The topic is interesting and worthy of note, however, the manuscript have to be improved. The introduction and the methods are described decently, whereas the discussion needs extensive English language correction.

I will address some comments to the study:

  1. It is not appropriate to do statistics in one, the same group. It is wrong use of statistics for two-way tables. This kind of statistical analysis should be use for finding a statistic difference in at least two groups. At the beginning of paragraph 3.2, the authors calculated the significance of MRSA prevalence in tested group in comparison to MSSA prevalence (p=0.006 ???).
  2. In my opinion, Table 1 shows the raw data of the study. It should be deleted or put into supplementary data.
  3. The authors should remember about italics when write the name of bacterial species.
  4. lines 212-213: The sentence: " However, the percentage of MP contamination was lower than other studies [11-13] and similar to another study [40], so this may depend on several factors, especially environmental [42,43]" should be developed. It needs some information inside.
  5. line 210-211: "In this work, the contamination with S. aureus of MP of health science students was studied, it was found that, as in other studies, that there is contamination of MP." It is better to split it into two sentences.
  6. In general, the discussion is hard to read. Too long sentences create unclear message and difficulties to follow up.

Author Response

I appreciate your comments which make the manuscript better

An extensive revision of the English was done.

  1. It is not appropriate to do statistics in one, the same group. It is wrong use of statistics for two-way tables. This kind of statistical analysis should be use for finding a statistic difference in at least two groups. At the beginning of paragraph 3.2, the authors calculated the significance of MRSA prevalence in tested group in comparison to MSSA prevalence (p=0.006 ???).

To calculate this value, a Chi square was made using the SPSS program, p<0.0001 was found. The Table is not put, only the data is put.

MSSA

MRSA

Pharynx

60 (41.66%)

13 (9.02%)

Nose

40 (27.77%)

12 (8.33%)

Mobile Phone

10 (6.94%)

9 (6.25%)

Total

110(76.38%)***

34 (23.61)

  1. In my opinion, Table 1 shows the raw data of the study. It should be deleted or put into supplementary data.

I appreciate your comment, however one of the reviewers asked to leave Table 1 with changes.

  1. The authors should remember about italics when write the name of bacterial species.

The entire manuscript was revised to italicize the names of the bacteria.

  1. lines 212-213: The sentence: " However, the percentage of MP contamination was lower than other studies [11-13] and similar to another study [40], so this may depend on several factors, especially environmental [42,43]" should be developed. It needs some information inside.

           Phrases were corrected

The percentage of contamination with S. aureus of the MP of the group of health students that we analyzed was lower than the one registered by another study (77.8%) [11]. In contrast we only found a percentage of contamination of 9.5%: this finding is more consistent with other studies which registered lower contamination ranging from 3.4% to 16.2% [12,13,40]. These differences in the percentages of contamination might be due to various factors, especially environmental ones. Recent research demonstrates that the contamination percentage of MP is higher among healthcare workers, when compared to people who do not work in hospitals [42].  Furthermore, a greater incidence of contamination of MP has been found in studies carried out in developing countries, when compared to developed countries [43].

  1. line 210-211: "In this work, the contamination with S. aureus of MP of health science students was studied, it was found that, as in other studies, that there is contamination of MP." It is better to split it into two sentences.

Phrases were corrected.

In this work, the contamination with S. aureus of health sciences students’ MP was studied. In a manner consistent with similar studies, we found MP contaminated S. aureus [11,12,40,41]

  1. In general, the discussion is hard to read. Too long sentences create unclear message and difficulties to follow up.

Discussion rearranged

Reviewer 2 Report

The topic of this article is interesting and important, thus mobile phones are items of every day usage. Thus the role of mobile phone in the transmission of Staphylococcus aureus is an important point to study. Despite these positive issues I have some quite important observations dominantly for the presentation of results, which seem to me to be quite not providing an easy survey and requiring a deeper discussion.

I have these main following observations to the submitted article.

Point 1

I miss the specification of control strains collections, from which the authors provided the control strains described as BAA-XXX or NRS-XXX (l. 96-97, l. 117-118).

Point 2

In chapter 3.1. Detection of Staphylococcus aureus carriers the presentation of data starts with the percentages of S. aureus (SA) carriers ranked by the human body part from which SA were isolated (pharynx, nose, pharynx and nose). Then it continues with grouping these percentages by gender, but not only for the human body part, but also it jumped for mobile phone, despite their contamination is given later. At first I prefer these percantages (women, men, all/positive isolation of SA from nose, pharynx, nose and pharynx, mobile phone) to be presented in a table. At second it would be advisable to add the statistical analysis, whether there was a statistically significant differences for the isolation of SA from nose, pharynx, nose and pharynx, and mobile phone between women and men. The occurrence in human body parts as pharynx and nose might be very similar for both genders. But as both genders might differ in some aspects of skin hygiene (e.g. using more cosmetics or make-up by women or using after shave lotion by men), it could be interesting to see, whether it may effect the transmission of SA on mobile phone (e.g. by antimicrobial effect of used cosmetics). Other statistical analysis should be focused whether the positive isolation of SA from different human body parts (nose, pharynx, nose and pharynx) influences significantly the presence of SA on a mobile phone. It might be that SA strains from nose or pharynx may differ in their phenotypic features as biofilm formation due to possible different conditions in pharynx and nose. This fact is mentioned between lines in more places, but in chapter 3.1 it should be pointed.

Point 3

In chapter 3.2. Characterization of isolated Staphylococcus aureus strains they are presented in Figure 1 the PFGE patterns for 47 strains, studied in detail, together with a first part of their features. One of the main hypotheses of the article is to study which factors influence the transmission of SA in single carriers, another is the diversity of SA within one carrier and within the whole group of students. But these relations are not visible very clearly from Figure 1. If I see the PFGE patterns, I am interested to see easily, which carriers hosted the identical SA strains in their body parts and mobile phone and which carriers hosted different SA strains. E.g. 34M and 34N seem to be identical (except biofilm formation), while 34P differ. On the other hand all isolates of student 2 as 2M, 2N and 2P seem to be completely identical. But to discover this it is necessary to check the whole table. I emphasize “to seem” on purpose, that the second part of genotyping features is presented (much more clearly) in Table 1. I see more ways how to make results more well-arranged. But it seems to me the most logical to transfer the data about biofilm formation from Figure 1 to Table 1 (as here are data on adhesin genes). By this it will be visible in Figure 1 that the strains identical by PFGE pattern are also identical in (in Figure 1 presented features) MIC, spa-type, mecA, PVL, SCCmec, strain type and these columns may be merged for strains with the PFGE identical pattern. Then the situation when all strains from one carier are identical, may be easily highlighted by the same colour (or to highlight the opposite situation, whether strains from one carrier belong to different genotype). To involve the information from Table 1, the asterisk for marking strains genotypically identical by Figure 1, but differing in toxin and adhesin genes, may be used.

Point 4

In Table 1 I am not sure it is necessary to involve the columns for toxin genes see and etb as they were not detected in any of isolates. This might be mentioned only in the text. It will enable to involve a column for biofilm formation and another column for marking, which strains were genotypically identical by Figure 1. This is very important to me to see which strains in one carrier were identical or not (see Point 5 too).

Point 5

For Table 1 I am slightly surprised how many strains genotypically identical by Figure 1 (PFGE pattern and other features) differ in toxin and adhesin genes presence. Could you please recheck the appropriate PCR results at first ? For adhesin genes it is explained in the discussion by comparing with the study of Noumi et al. [52]. But in this study only SA strains isolated students´ mobile phones were analysed. Thus the affirmation that the presence of toxin and adhesin genes depends on the place of isolation (pharynx, nose, mobile phone) requires deeper discussion. Do you suppose that by transferring to mobile phone some these adhesin and toxin genes are easily lost (if yes, it is genetically possible ? on which part of genomes are these genes encoded ?). Or do you suppose that strains possesing or not possesing some adhesin and toxin genes are more often transferred to mobile phone (or different parts of body) ? Then it is a question why the same strains (with or without these genes) were not isolated also from the human body. I would expect that the main place for evolving and shaping SA genome is human body, on which SA strains live and grow and from which they are transferred to a mobile phone, on which they only survive (without other intensive regrowth or evolving). Or are there some studies describing SA strains are able to grow and evolve on a mobile phone ?

Point 6

If you add the information about the biofilm formation in Table 1, please discuss whether their is the relation among adhesin genes and the biofilm formation.

Author Response

I appreciate your comments which make the manuscript better

 An extensive revision of the English was done.

Point 1

I miss the specification of control strains collections, from which the authors provided the control strains described as BAA-XXX or NRS-XXX (l. 96-97, l. 117-118).

The collections from which the strains come were indicated.

The following S. aureus strains from the ATCC (BAA strains) and the Network of Antimicrobial Resistance in Staphylococcus aureus (NRS strains) collections were used as positive controls:

Point 2

In chapter 3.1. Detection of Staphylococcus aureus carriers the presentation of data starts with the percentages of S. aureus (SA) carriers ranked by the human body part from which SA were isolated (pharynx, nose, pharynx and nose). Then it continues with grouping these percentages by gender, but not only for the human body part, but also it jumped for mobile phone, despite their contamination is given later. At first I prefer these percantages (women, men, all/positive isolation of SA from nose, pharynx, nose and pharynx, mobile phone) to be presented in a table. At second it would be advisable to add the statistical analysis, whether there was a statistically significant differences for the isolation of SA from nose, pharynx, nose and pharynx, and mobile phone between women and men. The occurrence in human body parts as pharynx and nose might be very similar for both genders. But as both genders might differ in some aspects of skin hygiene (e.g. using more cosmetics or make-up by women or using after shave lotion by men), it could be interesting to see, whether it may effect the transmission of SA on mobile phone (e.g. by antimicrobial effect of used cosmetics). Other statistical analysis should be focused whether the positive isolation of SA from different human body parts (nose, pharynx, nose and pharynx) influences significantly the presence of SA on a mobile phone. It might be that SA strains from nose or pharynx may differ in their phenotypic features as biofilm formation due to possible different conditions in pharynx and nose. This fact is mentioned between lines in more places, but in chapter 3.1 it should be pointed.

 The Table was attached and the statistical analysis was performed:

When grouped by gender, analysis showed that 71.64% (48 of 67) of men and 43.6% (58 of 133) of women carried S. aureus at one or more of the sites analyzed (p = 0.008). However, when the presence of S. aureus was weighed by comparing the gender and the different sites separately, no statistically significant differences were found between men and women. (Table 1).

Table 1. Distribution of Staphylococcus aureus strains in the different niches by gender.

No carriers

Carriers

Pharynx

Nose

Pharynx and nose

Mobile phone

Men

19 (9.5%)

48 (24%)

18 (16.98%)

16 (15.09%)

12 (11.76%)

7 (36.84%)

Women

75 (37.5%)

58 (29%)

33 (32.35%)

14 (13.72%)

11 (10.78%)

12 (63.15%)

Total

94 (47%)

106 (53%)

51 (48.11%)

30 (28.3%)

25 (23.58%)

19 (100%)

Point 3

In chapter 3.2. Characterization of isolated Staphylococcus aureus strains they are presented in Figure 1 the PFGE patterns for 47 strains, studied in detail, together with a first part of their features. One of the main hypotheses of the article is to study which factors influence the transmission of SA in single carriers, another is the diversity of SA within one carrier and within the whole group of students. But these relations are not visible very clearly from Figure 1. If I see the PFGE patterns, I am interested to see easily, which carriers hosted the identical SA strains in their body parts and mobile phone and which carriers hosted different SA strains. E.g. 34M and 34N seem to be identical (except biofilm formation), while 34P differ. On the other hand all isolates of student 2 as 2M, 2N and 2P seem to be completely identical. But to discover this it is necessary to check the whole table. I emphasize “to seem” on purpose, that the second part of genotyping features is presented (much more clearly) in Table 1. I see more ways how to make results more well-arranged. But it seems to me the most logical to transfer the data about biofilm formation from Figure 1 to Table 1 (as here are data on adhesin genes). By this it will be visible in Figure 1 that the strains identical by PFGE pattern are also identical in (in Figure 1 presented features) MIC, spa-type, mecA, PVL, SCCmec, strain type and these columns may be merged for strains with the PFGE identical pattern. Then the situation when all strains from one carier are identical, may be easily highlighted by the same colour (or to highlight the opposite situation, whether strains from one carrier belong to different genotype). To involve the information from Table 1, the asterisk for marking strains genotypically identical by Figure 1, but differing in toxin and adhesin genes, may be used.

Results from the Biofilm Formation column were transferred from Figure 1 to the current Table 2 (formerly Table 1) and identical strains were highlighted. In Figure 1, the cells of the strains with an identical pattern of PFGE were highlighted in blue. Likewise, in Table 2, the strains that are identical by PFGE and spa-type are highlighted in blue.

Blue cells: identical strains by PFGE

Point 4

In Table 1 I am not sure it is necessary to involve the columns for toxin genes see and etb as they were not detected in any of isolates. This might be mentioned only in the text. It will enable to involve a column for biofilm formation and another column for marking, which strains were genotypically identical by Figure 1. This is very important to me to see which strains in one carrier were identical or not (see Point 5 too).

The see and etb column was eliminated and it is only mentioned in the text and in Table 2, that no strains were found that presented these genes. The biofilm formation column was added and the strains that are identical by genotype were highlighted.

Point 5

For Table 1 I am slightly surprised how many strains genotypically identical by Figure 1 (PFGE pattern and other features) differ in toxin and adhesin genes presence. Could you please recheck the appropriate PCR results at first ? For adhesin genes it is explained in the discussion by comparing with the study of Noumi et al. [52]. But in this study only SA strains isolated students´ mobile phones were analysed. Thus the affirmation that the presence of toxin and adhesin genes depends on the place of isolation (pharynx, nose, mobile phone) requires deeper discussion. Do you suppose that by transferring to mobile phone some these adhesin and toxin genes are easily lost (if yes, it is genetically possible ? on which part of genomes are these genes encoded ?). Or do you suppose that strains possesing or not possesing some adhesin and toxin genes are more often transferred to mobile phone (or different parts of body) ? Then it is a question why the same strains (with or without these genes) were not isolated also from the human body. I would expect that the main place for evolving and shaping SA genome is human body, on which SA strains live and grow and from which they are transferred to a mobile phone, on which they only survive (without other intensive regrowth or evolving). Or are there some studies describing SA strains are able to grow and evolve on a mobile phone ?

The results of the PCR were reviewed and indeed some errors were found that were changed.

For us it is also interesting to find that strains typified as identical, have different genotypes, it is not the first time that we find this difference in the genes of toxins and adhesins of strains typified as identical, you can see the following articles:

Hamdan-Partida A, González-García S, de la Rosa García E, Bustos-Martínez J. 2018. Community-acquired methicillin-resistant Staphylococcus aureus can persist in the throat. International Journal of Medical Microbiology. 308(4):469-475. ISSN: 1438-4221. https://doi.org/10.1016/j.ijmm.2018.04.002

These can also be reviewed:

Genotyping of Staphylococcus aureus strains isolated from healthy persistent carriers. Folia Microbiol (2014) 59:349–353

Dynamic pattern and genotypic diversity of Staphylococcus aureus nasopharyngeal carriage in healthy pre-school children. J Antimicrob Chemother 2013; 68: 1517–1523 doi:10.1093/jac/dkt080

We have found in another study differences in the genotypes of nose and pharynx strains.

Hamdan-Partida A, Sainz-Espuñes T, Bustos-Martínez J. 2010. Characterization and persistence of Staphylococcus aureus isolated from the anterior nares and throat from healthy carriers in a Mexican community. Journal of Clinical Microbiology. 48(5):1701-1705. doi: 10.1128/JCM.01929-09. ISSN: 1201-9712

What we can say is that genetic variability is very wide in S. aureus strains and we are precisely investigating whether there are changes depending on the niche studied by means of complete genome sequencing, to better visualize these changes.

We do not believe that genes are easily lost when passing from humans to cell phones, nor that S. aureus evolves more easily in cell phones, there is no report to our knowledge that supports this fact.

Point 6

If you add the information about the biofilm formation in Table 1, please discuss whether their is the relation among adhesin genes and the biofilm formation.

The discussion in relation to adhesins and the formation of biofilms was enlarged

Reviewer 3 Report

General considerations: In the manuscript entitled “Molecular characterization of Staphylococcus aureus strains isolated from mobile phones”, the authors analyzed the possible contamination of 200 mobile phones from university health science students with Staphylococcus aureus strains. Since the aim of the study was to determine the source of contamination of the mobile phones, paired nasal and throat swabs were also collected from these students. S. aureus strains were isolated from 19 mobile phones, and an association was established with S. aureus strains isolated from the nose and/or pharynx of their owners. Typing of S. aureus isolates was performed using PFGE and spa-typing, the presence of adhesin and toxin genes was evaluated, and the ability to form biofilms was determined. An association was found between the S. aureus strains isolated from mobile phones and the strains isolated from the nose and/or pharynx of their owners, suggesting that the mobile phones can be contaminated with S. aureus strains that colonize their owners, including MRSA strains. Despite the importance of the results reported in the manuscript, I strongly recommend revising the paper to improve the contextualization and the discussion of the results.

Major Comments

Line 46-51: It is not clear why did you affirm “S. aureus has been seen to colonize mainly the nose”, and then claim “However, the nose does not appear to be the main niche that S. aureus can colonize”. Could you please explain more clearly this idea?

Line 48: In your introduction you mention that “the hands and the nose can be a source of MP contamination”, and people are constantly manipulating their mobile phones with their bare hands.  Why, in addition to the nose and pharynx samples, didn't you do a hand sample collection?

Line 52: “S. aureus colonizes using adhesins, proteins with which it binds to surfaces.”. S. aureus uses multiple adhesins to establish colonization in all its reservoirs, or is there any specificity?

Line 52-56: Despite mentioning the general importance of adhesins and toxins for the colonization and infection of S. aureus, it would be important to mention and briefly contextualize, at least, the toxins and adhesins selected, and after detected in the isolated S. aureus strains.  

Line 133-134; 140-141 and 230-231: In the results section, you mention that “18.45% of the carrier students presented contamination in their MP in addition to the nose and/or pharynx” and that “only 19 mobile phones (9.5%) of the total were found to be contaminated with S. aureus”. Can you please explain why in the discussion you mention: “Although there are studies that indicate that MP are not significant reserves of MRSA strains [46], our results indicate the opposite, “?

Discussion: The discussion is not easily understandable. Instead of just mentioning “other study”, “lower than other studies and similar to another study”, could you please mention some relevant details of the study and how it can be compared to this study?  

Minor Comments:

Line 35: MRSA strains are a remarkable public health problem, but please review the sentence “Methicillin- resistant strains of S. aureus (MRSA) are the most pathogenic…strains”.

Line 40: Please explain the acronym “ICU”.

Line 43: “mobility of strains from the hospital to the community and vice versa from the community to the hospital”. As the idea is repeated, please change it to: “mobility of strains from the hospital to the community and vice versa.” or “mobility of strains from the hospital to the community and from the community to the hospital”

Line 85 and 86: “PCR assays were performed for mecA gene using primers and conditions for mecA were as reported previously”, please replace by “PCR assays were performed for mecA gene using primers and conditions as reported previously”

Line 159: In the sentence “SCCmec type was detected only when the strain had MIC ≥4 μg/mL.”, do you mean “was tested” instead of “was detected”?

Line 170, 184: Please italicize “S. aureus” and “Staphylococcus aureus”. Please confirm that species names are all italicized in the manuscript.

Author Response

I appreciate your comments which make the manuscript better

An extensive revision of the English was done.

Major Comments

Line 46-51: It is not clear why did you affirm “S. aureus has been seen to colonize mainly the nose”, and then claim “However, the nose does not appear to be the main niche that S. aureus can colonize”. Could you please explain more clearly this idea?

The phrase was changed

For a long time, the nose has been considered as the main ecological niche of S. aureus is [15]. However, there is evidence that it can colonize the skin, axillae, groin, rectum, hands, and pharynx [16-19]. Furthermore, recent studies demonstrate that the colonization of the pharynx by S. aureus can be greater than in the nose, which puts into question that the latter is the main colonization niche [19-21]. Therefore, nose and pharynx can also be a source of S. aureus MP contamination.

Line 48: In your introduction you mention that “the hands and the nose can be a source of MP contamination”, and people are constantly manipulating their mobile phones with their bare hands.  Why, in addition to the nose and pharynx samples, didn't you do a hand sample collection?

When the samples were taken, the hands were not considered since we wanted to focus mainly on the pharynx, which is the niche that our group studies. Seeing the results obtained, we realize that hand sampling is necessary to have a more complete picture. Therefore, future studies will also take into account.

Line 52: “S. aureus colonizes using adhesins, proteins with which it binds to surfaces.”. S. aureus uses multiple adhesins to establish colonization in all its reservoirs, or is there any specificity?

What has been seen is that S. aureus has a series of adhesins that serve to establish itself on all surfaces, apparently they do not have a certain specificity. However, this has not yet been fully demonstrated, so we are studying whether there are differences between the colonization factors of the nose and pharynx, including adhesins.

Line 52-56: Despite mentioning the general importance of adhesins and toxins for the colonization and infection of S. aureus, it would be important to mention and briefly contextualize, at least, the toxins and adhesins selected, and after detected in the isolated S. aureus strains. 

Added in the discussion the importance of adhesins and toxins

Line 133-134; 140-141 and 230-231: In the results section, you mention that “18.45% of the carrier students presented contamination in their MP in addition to the nose and/or pharynx” and that “only 19 mobile phones (9.5%) of the total were found to be contaminated with S. aureus”. Can you please explain why in the discussion you mention: “Although there are studies that indicate that MP are not significant reserves of MRSA strains [46], our results indicate the opposite, “?

The sentence was changed

.  Brady et al, suggest that there is a low possibility of MP contamination with MRSA strains in non-hospital environments [46]. Our results show that it is possible that MP are contaminated with MRSA strains (Figure 1), even though students are not in hospital settings or environments. Fortunately, the percentage of MRSA strains is still lower than that of MSSA strains found by other studies, that also show low contamination with MRSA strains

Discussion: The discussion is not easily understandable. Instead of just mentioning “other study”, “lower than other studies and similar to another study”, could you please mention some relevant details of the study and how it can be compared to this study? 

Discussion was rearranged.

Minor Comments:

Line 35: MRSA strains are a remarkable public health problem, but please review the sentence “Methicillin- resistant strains of S. aureus (MRSA) are the most pathogenic…strains”.

The phrase was changed

Thus, S. aureus represents a major public health problem, especially in the case of Methicillin-resistant S. aureus (MRSA) strains, which are more pathogenic.

Line 40: Please explain the acronym “ICU”.

Full name added

Intensive Care Unit

Line 43: “mobility of strains from the hospital to the community and vice versa from the community to the hospital”. As the idea is repeated, please change it to: “mobility of strains from the hospital to the community and vice versa.” or “mobility of strains from the hospital to the community and from the community to the hospital”

The phrase was changed.

Line 85 and 86: “PCR assays were performed for mecA gene using primers and conditions for mecA were as reported previously”, please replace by “PCR assays were performed for mecA gene using primers and conditions as reported previously”

The phrase was changed.

Line 159: In the sentence “SCCmec type was detected only when the strain had MIC ≥4 μg/mL.”, do you mean “was tested” instead of “was detected”?

The word was changed

Line 170, 184: Please italicize “S. aureus” and “Staphylococcus aureus”. Please confirm that species names are all italicized in the manuscript.

It was revised in all the text and it was written correctly

Round 2

Reviewer 2 Report

I really appreciate correcting and improving this article and the work performer for this study. Regarding this I agree with the performed changes and I have one minor point regarding Table 1 and a few formal points summarised in point 2 to be considered before this article publishing.

I have two comments for future research. At first to understand which factors influences the transferring of S. aureus isolates to mobile phones, it would be also useful to study in future, whether they are some dominant features in strains not to be transferred to mobile phones (and to be isolated only from carriers). At second some other genotyping methods as MLST or coreMLST or wgMLST sequencing are now considered for a deep comparison of isolates with the same PFGE profile as long-time colonising strains can develop during time.

Point 1

I appreciate Table 1, which gives the clear demonstration of the ocurrence within single subgroups. I have only a comment for the arrangement. In Table 1 the number of positive carriers within single groups (men, women, total) is given. What I do not think to be very well explained is the percentages data. These percentages are referred to different bases. Total + No carriers and Carries is referred to the whole group (200), total + pharynx, nose and pharynx nose to the positive participants (94) and total + mobile phone itself is taken as 100 %. To demonstrate all data in a organized way please consider to refer percentages for total + mobile phone agains the whole group (200) and to add another column for total + mobile phone + carriers (which should be 10 instead of 19).

For men and women data on percentages I consider it is more important to demonstrate which was the prevalence in these subgroups thus the percentages I consider to be referred to the single subgroup itself. It also illustrates better the observations given in l. 162-166.  In contrast now the percentages are expressed as the percentages of men and women in singl carriers´ groups, which is negligble as these groups are of different size. As e.g. for „pharynx“ it will be: men – 18 (27 %), women – 33 (25 %), total – 26 %. Regarding the frequency of positivity within men and women please specify which level for the statistical significance and which test you used. As e.g. the probability for positive isolation from pharynx was 27 % for men and 24 % for women, for positive isolation from nose it was 24 % for men, but only 10.5 % for women. And the way of percentages should be also explained in the table description.

Point 2

In line 189 „S.“ misses and I suggest to add information about the whole amount of isolates from mobile phones and to correct as „S. aureus found in MP (19 isolates), PFGE established….“. In Figure 2 please consider to use different colours for genes clfa and cna, as the used blue colour seems to be the same completely and it is difficult to be distinguished. In the discussion (l. 284-285) (Recent research demonstrates that the contamination percentage of MP is higher among healthcare workers, when compared to people who do not work in hospitals [42]) please emphasize whether  the group of health science students worked also in the hospital or not.

Author Response

I appreciate your comments

I really appreciate correcting and improving this article and the work performer for this study. Regarding this I agree with the performed changes and I have one minor point regarding Table 1 and a few formal points summarised in point 2 to be considered before this article publishing.

I have two comments for future research. At first to understand which factors influences the transferring of S. aureus isolates to mobile phones, it would be also useful to study in future, whether they are some dominant features in strains not to be transferred to mobile phones (and to be isolated only from carriers). At second some other genotyping methods as MLST or coreMLST or wgMLST sequencing are now considered for a deep comparison of isolates with the same PFGE profile as long-time colonising strains can develop during time.

I appreciate your comments and we will take them into account for future research

Point 1

I appreciate Table 1, which gives the clear demonstration of the ocurrence within single subgroups. I have only a comment for the arrangement. In Table 1 the number of positive carriers within single groups (men, women, total) is given. What I do not think to be very well explained is the percentages data. These percentages are referred to different bases. Total + No carriers and Carries is referred to the whole group (200), total + pharynx, nose and pharynx nose to the positive participants (94) and total + mobile phone itself is taken as 100 %. To demonstrate all data in a organized way please consider to refer percentages for total + mobile phone agains the whole group (200) and to add another column for total + mobile phone + carriers (which should be 10 instead of 19).

The MP column was changed against the total sample (200).

Regarding the suggested column of total+MP+carriers, I must clarify that only one student presented contamination of his MP without being a carrier in the nose and/or pharynx. Therefore, the remaining 18 students did present S. aureus in the nose and/or pharynx, and MP. Of these, 10 present the same MP strain in the nose and/or pharynx and 8 did not present the same strain in any of the niches. So, we think it is not possible to put the suggested column in the same format as Table 1. For what is explained in Lines 158 to 162.

For men and women data on percentages I consider it is more important to demonstrate which was the prevalence in these subgroups thus the percentages I consider to be referred to the single subgroup itself. It also illustrates better the observations given in l. 162-166.  In contrast now the percentages are expressed as the percentages of men and women in singl carriers´ groups, which is negligble as these groups are of different size. As e.g. for „pharynx“ it will be: men – 18 (27 %), women – 33 (25 %), total – 26 %. Regarding the frequency of positivity within men and women please specify which level for the statistical significance and which test you used. As e.g. the probability for positive isolation from pharynx was 27 % for men and 24 % for women, for positive isolation from nose it was 24 % for men, but only 10.5 % for women. And the way of percentages should be also explained in the table description.

The following phrases were added:

However, when the prevalence of S. aureus was analyzed comparing by gender and isolation sites, we found that for the pharynx, men presented a prevalence of 26.86% and 24.81% for women. In the case of the nose, the prevalence in men was 23.88% and in women 10.38%. Finally, the prevalence for men and women in both sites was 17.91% and 8.27% respectively. No statistically significant differences were found between men and women in any of the cases (p >0.05).

Line 135-138.  For the statistical analysis of the prevalence according to gender, a multiple comparison was made using a two-way ANOVA followed by Tuckey's test with GraphPad Prism software version 8.0 for Windows

Point 2

In line 189 „S.“ misses and I suggest to add information about the whole amount of isolates from mobile phones and to correct as „S. aureus found in MP (19 isolates), PFGE established….“.

Phrase corrected

Of the strains of S. aureus found in MP (19 isolates)

In Figure 2 please consider to use different colours for genes clfa and cna, as the used blue colour seems to be the same completely and it is difficult to be distinguished.

Colors were changed

In the discussion (l. 284-285) (Recent research demonstrates that the contamination percentage of MP is higher among healthcare workers, when compared to people who do not work in hospitals [42]) please emphasize whether  the group of health science students worked also in the hospital or not.

These two phrases were added to emphasize that students do not work in hospitals

Line 64. of the first year of the degree, still without going to hospital centers

Line 267.  as is the case of the students analyzed in this work.

Reviewer 3 Report

Dear authors,

Thank you for the revision of the manuscript.  I am satisfied with the clarifications and the revision made, and I recommend it for publication. 

Minor comments:

Line 188-189: It seems that the beginning of the sentence is missing.

Line 223: Please italicize "icaA"

Author Response

I appreciate your comments

Minor comments:

Line 188-189: It seems that the beginning of the sentence is missing.

The lines of the manuscript were composed

Line 223: Please italicize "icaA"

The word was italicized